# Developing the Mental Health Sensitive School Concept: Insights from focus group discussions

**Viktoriia Gorbunova**[1,2]**, Vitalii Klymchuk**[ID][2,3]*****, Olha Faryma**[ID][4,5]

**1** Department of Social and Applied Psychology, Zhytomyr Ivan Franko State University, Zhytomyr, Ukraine, **2** Mental Health for Ukraine Project, Lviv, Ukraine, **3** National Psychological Association of Ukraine, Kyiv, Ukraine, **4** NGO Be Healthy, Kyiv, Ukraine, **5** Bern University of Applied Sciences, Bern, Switzerland

* vitaly.klymchuk@gmail.com

## Abstract

The ongoing war has exacerbated mental health challenges among students and educators in Ukraine, placing increased strain on the psychological support system within schools. While school-based psychological services have been integrated into Ukraine's education system since 1993, they remain underfunded, understaffed, and structurally isolated from broader school policies and external healthcare services (also, with no clear focus on mental health). This study explores the development of a Mental Health-Sensitive School Concept (MHSS), which seeks to integrate mental health considerations into educational environments using a systemic and participatory approach. A qualitative research design was employed, using 12 FGDs with 205 participants across five Ukrainian regions. The participant pool included school administrators, teachers, psychologists, social pedagogues, and students. Discussions focused on mental health awareness, intervention feasibility, implementation barriers, and stakeholder roles. Thematic analysis was applied to synthesise key findings. Stakeholders strongly supported integrating mental health initiatives within school policies but highlighted critical challenges such as limited resources, staff burnout, stigma, and a lack of structured mental health training for teachers. Key thematic areas identified included mental health awareness and stigma reduction, school-based interventions, policy development, capacity-building, and adaptation to crisis contexts. The study emphasised the need for structured policy support, expanded psychological services, and improved teacher training to foster a sustainable and trauma-informed MHSS framework. The findings underscore the urgent need for systemic reforms to embed mental health considerations into school environments. The study proposes a tiered intervention model encompassing mental health promotion, prevention, early psychological support, and structured referral mechanisms. Successful implementation will require policy endorsement, financial investment, and multi-stakeholder collaboration. Future research should evaluate the long-term impact of the MHSS on student and educator well-being.

**Data availability statement:** Data supporting these findings are available in Open Access in Zenodo (Klymchuk, V., Gorbunova, V., & Faryma, O. (2025). Mental Health Sensitive School Concept: Focus Group Discussions [Data set]. Zenodo. https://doi.org/10.5281/zenodo.14922811)

**Funding:** The study was conducted with the support of the "Mental Health for Ukraine Project" (MH4U), implemented in Ukraine by GFA Consulting Group GmbH and funded by Switzerland through the Swiss Agency for Development and Cooperation (SDC). The project aims to improve mental health outcomes among the Ukrainian population. The sponsors were not involved in the study design, collection, analysis and interpretation of the data or preparation of the article. No funds were dedicated to the article preparation.

**Competing interests:** The authors have declared that no competing interests exist.

## Introduction

The mental health support gap in Ukraine is increasing due to many factors, including the ongoing war initiated by Russia, which escalated into a full-scale invasion in 2022 [1,2]. Children, one of the vulnerable groups, are among the most affected during such crises [3].

Due to their unique position in the communities, educational facilities play a crucial role in supporting children`s well-being, including mental health. The psychological support system has been integrated into the Ukrainian education system since 1993 (*here and further, the meaning of "psychological support" includes a broad definition as support of "comprehensive development of individuals, protecting their mental and physical health, and assisting in forming a harmonious educational environment"; the meaning of "mental health support" includes all support and interventions provided specifically for mental health promotion, prevention, treatment and rehabilitation*) [4]. The Law of Ukraine "On Education" (Law No. 2145-VIII), enacted on September 5, 2017, acknowledges the importance of psychological services within the educational context. According to Article 76, the psychological service in the education system is designed to promote the comprehensive development of individuals, protect their mental and physical health, and assist in forming a harmonious educational environment. The service aims to provide psychological and socio-pedagogical support to participants in the educational process, including students, educators, and parents [5].

While the law outlines the general purpose and significance of psychological services, practical psychologists' specific tasks and responsibilities are detailed in subordinate regulations, such as the Regulation on the Psychological Service in the Education System of Ukraine [6]. This regulation outlines the key tasks of school psychologists as follows: Psychological Diagnostics, Correctional and Developmental Work, Counseling, Preventive Activities, Educational and Methodological Support, Research Activities, and Organizational and Methodological Work.

The recently available data (2022/2023 academic year) shows that the psychological service in the Ukrainian education system consists of 20,413 specialists, including 13,174 psychologists, 6,905 social pedagogues, and 82 methodologists [7]. These professionals work in various educational institutions, including preschools, general secondary schools, vocational institutions, and higher education establishments. Despite its essential role, the psychological service in Ukraine faces several challenges [7]:

- Staff Shortages: There is a deficit of specialists, especially in rural areas, due to unattractive working conditions, low salaries, and wartime emigration.

- Insufficient Funding: Many regions lack adequate financial support for psychological services, limiting resources for mental health initiatives.

- Overload of Specialists: The high number of students per psychologist or social pedagogue results in heavy workloads and limited individual support.

- Impact of War: The ongoing war has led to displacement, infrastructure damage, and increased demand for trauma-focused interventions.

- Lack of Institutional Support: Many schools lack dedicated positions for psychologists in their staffing structure, making service delivery inconsistent.

- Limited Professional Development: While training is required, many specialists lack opportunities for continuous professional development due to financial and logistical constraints.

However, the main challenge preventing the system from further development is rooted in the legislation discourse. This discourse positions the mental health and children`s well-being of school psychologists solely as the responsibility of school psychologists without imposing any obligation on other school administrations or teachers. Hence, there is an overload, a lack of support and funding, and other challenges.

Several approaches were explored to solve and overcome this discourse challenge: the whole-school approach to mental health in general [8,9] and trauma-sensitive schools in particular [10].

The whole-school approach to mental health integrates well-being into all aspects of school life through key principles such as leadership and management, embedding mental health into school policies, curriculum, and teaching to promote resilience. Students' voices are heard, staff well-being and training ensure adequate support, and identifying needs and monitoring impact help assess and improve interventions [8].

A trauma-sensitive school is an educational environment that recognises the profound impact of trauma on students' learning, behaviour, and emotional well-being. Children affected by trauma often experience difficulties with emotional regulation, concentration, relationships, and trust in authority figures, which can hinder their ability to engage in learning. Trauma-sensitive schools adopt a whole-school approach to creating a safe, supportive, and inclusive learning environment. Key principles of trauma-sensitive schools include fostering physical and emotional safety, building strong student-teacher relationships, using restorative and strengths-based approaches, and ensuring collaborative decision-making between educators, families, and mental health professionals [10].

Based on these approaches and following the WHO HAT Guideline [11] the idea for the Mental Health Sensitive School Concept (MHSS) evolved. The MHSS is a structured framework that fosters a school environment that actively promotes mental well-being and supports students, teachers, and school staff. Rooted in international best practices and evidence-based interventions, this concept promotes integrating mental health into the education system through promotion, prevention, early intervention, and referral mechanisms.

The primary objective of the MHSS is to create mentally healthy, inclusive, and supportive school environments where students and staff feel safe, valued, and empowered. To achieve this, the concept emphasises **key goals**, including promoting mental health awareness, preventing psychological disorders, providing early support, and systematically referring for specialised care when needed. Additionally, it advocates for integrating mental health considerations into school policies and governance to ensure a long-term, sustainable impact [12]. *The detailed description of the MHSS in its current form is presented in the Discussion section and* Fig 1.

The study **aims** to develop and refine the Mental Health Sensitive School Concept as a structured framework that integrates mental health support into educational settings in Ukraine. By drawing on stakeholder perspectives through focus group discussions (FGDs), the research seeks to identify key principles, challenges, and practical implementation strategies for fostering mentally healthy, inclusive, and supportive school environments.

The primary **objective** of this study is to explore the perceptions, needs, and challenges of key stakeholders -including teachers, school administrators, psychologists, social pedagogues, and students - regarding integrating mental health support within schools. The study seeks to examine stakeholder perspectives on mental health-sensitive school environments, particularly regarding awareness, stigma reduction, and intervention feasibility; verify structural components for an MHSS framework, ensuring that it is addressing the specific challenges of the Ukrainian context; develop actionable

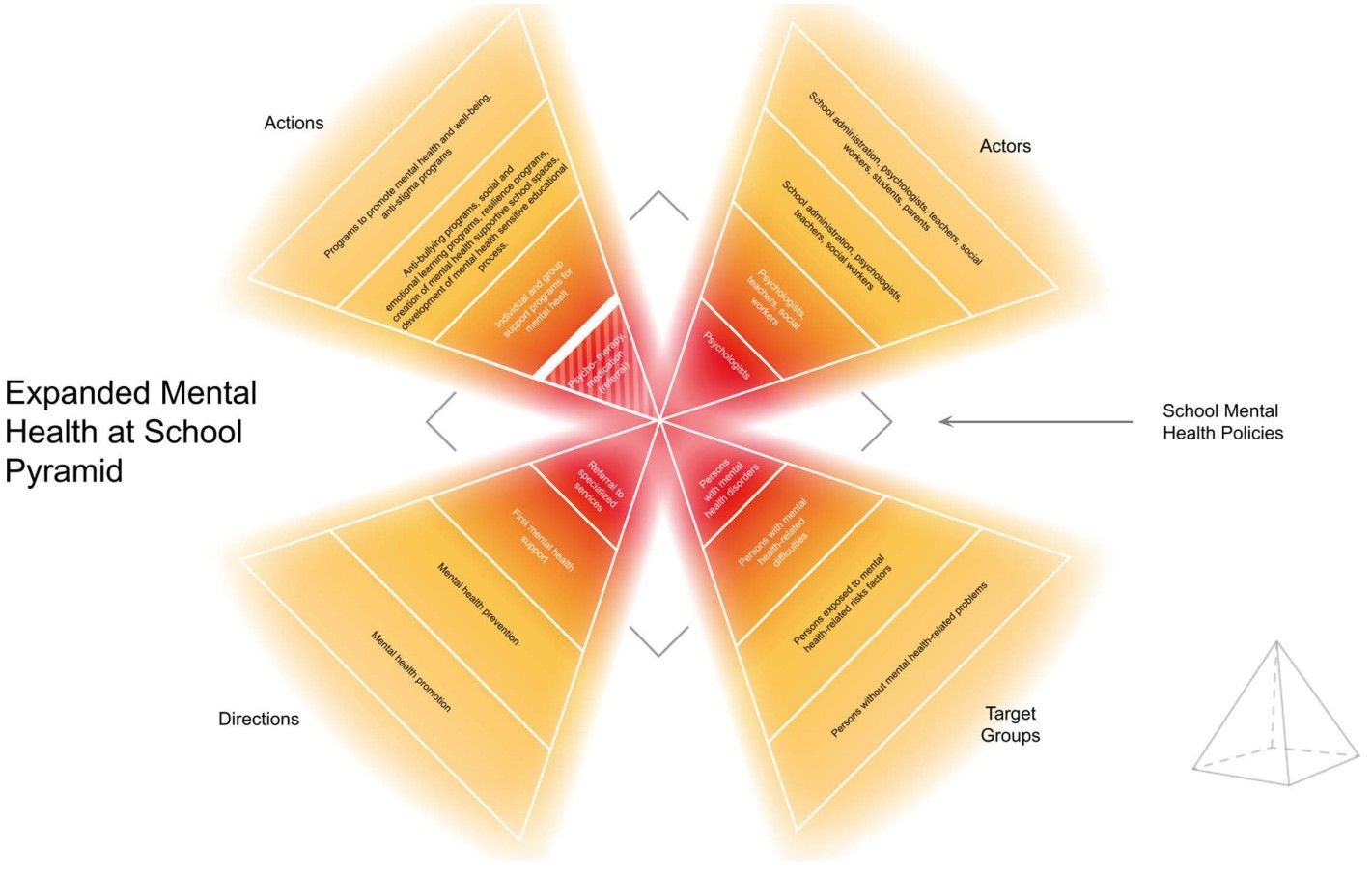

**Fig 1. Mental Health Sensitive School Pyramid.**

recommendations for the implementation of MHSS, including policy integration, teacher training, student support initiatives, and crisis-adaptive mental health interventions.

This study report follows the Standards for Reporting Qualitative Research (SRQR guideline) [13].

## Materials and methods

### Qualitative approach and research paradigm

The study on developing the Mental Health-Sensitive School Concept was rooted in the interpretive paradigm [14], as it aimed to explore stakeholder perspectives and collaboratively develop the concept further. In terms of approaches, the study adopts a pragmatic approach [15], focusing on the practical development of a Mental Health-Sensitive School Concept through stakeholder insights gathered via focus group discussions.

### Researchers' characteristics and reflexivity

**Assumptions and preconceptions.** The research team consisted of experts in psychology, mental health, and public health with experience in quantitative and qualitative research. Two of the authors (VK, VG) have been deeply involved in the transformation of the mental health system in Ukraine since 2016, supporting the development of the National Mental Health Strategy/Concept [16], and OF – involved in public health promotion initiatives, fostering health and well-being

among school children and teachers through national educational projects. We acknowledged several assumptions that influenced our work: a) we assumed that integrating mental health sensitivity into the school environment is essential for fostering students' overall well-being, academic success, and long-term resilience; b) we assumed that meaningful engagement of teachers, students, parents, and school counsellors is necessary to develop a contextually relevant and feasible Mental Health Sensitive School Concept; c) we assumed that any proposed framework for mental health-sensitive schools must be tailored to local needs, values, and resources rather than adopting a one-size-fits-all model; d) we assumed that schools represent a crucial setting for early intervention and prevention of mental health challenges, particularly for children and adolescents who may otherwise lack access to mental health services; and finally, e) we assumed from the beginning that the current system of psychological support is not performing well its functions due to the over-broad focus on psychological support of the education process, and not focusing on the mental health support of students and school personnel.

Acknowledging these assumptions, we remained aware of potential biases and strived to ensure that our findings were grounded in stakeholder perspectives rather than preconceived frameworks. This study's iterative and participatory nature aims to critically examine and refine these assumptions based on the lived experiences and insights shared during focus group discussions.

**Awareness of researcher assumptions and bias.** While efforts were made to conduct and interpret this research transparently, we recognise that all qualitative inquiry is shaped by the positionality and assumptions of the research team. We acknowledge the following biases:

• Intervention-optimism bias: As part of a project aiming to promote MHSS implementation, the research team may have unconsciously favoured positive interpretations of stakeholder feedback or underemphasised risks.

• Cultural embeddedness: Although researchers were native to the context or worked closely with Ukrainian partners, conceptual frameworks and language choices may still reflect Western/global health biases.

• Sampling constraints: Individuals with lower digital access or more critical stances may have been underrepresented due to the online format and volunteer-based recruitment.

We have attempted to mitigate these biases by: (a) anonymising transcripts prior to coding; (b) applying multiple-coder triangulation during analysis; and (c) explicitly including dissenting perspectives in the Results section.

**Positionality and power dynamics.** Given that two authors (VK, VG) have been involved in mental health system transformation and policy development, and OF has experience in public health promotion, we recognise that our roles may have positioned us as experts in the field. While this expertise provided valuable insights, it also introduced a potential power imbalance—participants may have perceived us as authority figures, influencing how they expressed their views. To mitigate this imbalance, we adopted a facilitative rather than directive role, ensuring that participants felt empowered to share their experiences and perspectives without seeking approval or validation from researchers.

Our background in developing national mental health strategies and public health initiatives has accustomed us to system-level and policy-oriented thinking. However, this study required a shift towards grassroots perspectives, focusing on the experiences of teachers, students, school psychologists, and administration. To balance these perspectives, we actively encouraged bottom-up insights, recognising that practical implementation challenges and needs may differ significantly from policy-level assumptions.

Mental health remains a sensitive and sometimes stigmatised topic, particularly in school environments. Teachers may have different levels of comfort discussing mental health openly due to societal norms, institutional policies, or personal beliefs. Additionally, the ongoing transformation of Ukraine's mental health and education systems means that some participants may have had prior negative experiences, affecting their willingness to engage openly. To navigate these dynamics, we ensured confidentiality and a non-judgmental environment, reinforcing that all perspectives were valuable and that there were no 'wrong' answers.

**Interaction with participants.** Our interactions with participants played a crucial role in shaping the research process, influencing the depth of the discussions and the quality of insights gathered. As the study relied on focus group discussions (FGDs) to develop the Mental Health Sensitive School Concept, our engagement with participants was designed to foster open dialogue, trust, and collaborative knowledge production. Recognising that mental health is a sensitive topic, particularly in school settings, we prioritised creating a supportive and non-judgmental space where participants felt comfortable sharing their thoughts, concerns, and personal experiences. This was achieved by clarifying the purpose of the discussion at the outset and emphasising that there were no right or wrong answers, using participatory facilitation techniques to encourage all voices to be heard, and ensuring that both dominant and quieter participants had the opportunity to contribute.

As focus groups included diverse stakeholder groups, power dynamics influenced the discussions. We adapted our approach to managing these dynamics by carefully structuring group compositions (e.g., separating students from teachers where necessary) to reduce potential hierarchy-driven constraints, using neutral and inclusive language and avoiding technical jargon to ensure accessibility for all participants, encouraging mutual respect among participants; and reinforcing that different perspectives were valid and welcomed.

We took a reflexive approach throughout our engagement with participants, continuously assessing how our presence, facilitation style, and prior assumptions might shape the discussions. After each focus group, we used field notes and debriefing sessions to reflect on participant responses and our role in shaping the discourse. Additionally, we adapted questioning techniques based on participant needs, ensuring discussions remained organic, inclusive, and relevant to their lived realities.

**Methodological choices and reflexivity.** In designing this study, we adopted a qualitative interpretivist paradigm and pragmatic approach, prioritising stakeholder experiences, perceptions, collaborative knowledge generation, and application of the knowledge for advancing the Concept. Given the study's objective - to develop a Mental Health Sensitive School Concept - we selected focus group discussions as the primary data collection method. This choice was driven by the need to facilitate dialogue, capture diverse viewpoints, and encourage interaction among participants, allowing for the co-construction of ideas and shared understandings.

For data analysis, we employed thematic analysis, which allowed us to identify recurring themes and patterns in participants' discussions, move beyond surface-level responses to explore underlying assumptions and meanings, and compare perspectives across different stakeholder groups, ensuring a holistic and inclusive conceptualisation of a Mental Health Sensitive School.

We engaged in ongoing reflexivity throughout the research process, acknowledging how our backgrounds, positionality, and methodological choices influenced the study. Reflexivity was particularly important given our prior involvement in mental health policy, system transformation, and public health promotion.

## Context

The study was conducted in Ukraine in the context of uncertainty due to the ongoing war (since 2014), full-scale invasion (February 2022), attempts to reform the school education system "New Ukrainian School" [17], and to transform the system of psychological support in the education system [18].

The starting point for the concept development was the report on best practices in mental health support in schools, commissioned by the MH4U Project to Implemental Worldwide [19]. Based on the report, a group of experts developed the initial draft of the Mental Health-Sensitive School Concept (the final version is available in open access [12].

This draft was the reference point for the FGDs. It was sent to the participants in advance, asking them to familiarise themselves with the content and ideas and consider the problems and solutions beforehand.

## Sampling strategy

The feasibility of recruitment determined the sample size of 205 FGD participants from 5 regions of Ukraine. The recruitment period for this study was from 01.06.2022 to 01.08.2022.

*Eligibility criteria for the FGDs:*

- School staff or school students (8-11th grade).

- Confirmation of informed consent to participate in the FGDs.

**Recruitment geography.** Participants were recruited across five Ukrainian regions (oblasts): Lviv, Dnipropetrovsk, Rivne, Chernivtsi, and Kyiv.

The sampling was made via 1) Official invitations that were sent by regional Departments of Education to all secondary-school principals with study information sheets; 2) Project newsletter & social-media posts (Facebook, Telegram) circulated via the Mental Health for Ukraine (MH4U) network; 3) Snowball referrals: early registrants were encouraged to share the Google-Form link with eligible colleagues and peers.

## Ethical issues

The research team adhered to the Declaration of Helsinki and the National Psychological Association of Ukraine Ethical Regulation. All participants signed the Informed Consent form prior to participating in the FGDs. The study protocol, FGDs scenario, and informed consent forms were approved by the Ethics Committee of the Zhytomyr Ivan Franko State University, approval number №02–0103/2022 of 01 March 2022.

## Data collection methods and instruments

Focus group discussions (FGDs) were the primary data collection method to gather stakeholder insights. The study design incorporated structured facilitation, participant engagement techniques, and guided thematic discussions to ensure that diverse perspectives were captured. Focus groups were conducted online using the Zoom application. Each session lasted up to 90 minutes and followed a semi-structured format, allowing for guided discussions and open-ended participant contributions.

The focus groups followed a structured approach, broken into key phases:

1. Introduction and Participant Engagement (15 minutes). Participants introduced themselves and shared their current emotional state (e.g., using visual prompts or self-reflection techniques). The purpose of the discussion was clarified, reinforcing that feedback would shape the finalisation of the MHSS.

2. Concept Review and Prioritization (15 minutes). Participants were asked whether they had read the draft Concept and were invited to share general observations and gaps. Discussions focused on implementation feasibility, stakeholder responsibilities, and adaptation to Ukraine's current context.

3. Thematic Discussion on Core Components (45 minutes). Participants were engaged in whole-group discussions to prioritise key issues within the Concept. The discussions covered models of school-based mental health programs and their relevance to Ukraine; intervention levels (promotion, prevention, early support, referral) and their applicability in school settings; implementation challenges, risks, and enablers for mental health-sensitive policies in schools; identification of key target groups and strategies for reaching them effectively; existing programs, initiatives, and best practices in schools that align with MHSS principles; structural and policy-related aspects, including school-level governance, working groups, and integration into school regulations.

4. Reflection and Action Planning (15 minutes): Participants identified 1–3 concrete takeaways for their schools. Then, they conducted a commitment-setting exercise and outlined specific first steps for implementation, timelines, and responsible persons.

A semi-structured topic guide was developed to ensure consistency across FGDs while allowing for flexibility in discussion. It included icebreakers and participant engagement questions to create a comfortable discussion atmosphere; core

questions exploring the relevance, feasibility, and gaps in the MHSSC draft; probing questions to elicit deeper insights, including open-ended prompts such as: *"What does a mental health-sensitive school look like in your context?", "What are the biggest barriers to implementing these interventions in your school?" "Which examples or models seem most applicable to Ukraine?"* (Topic guide is available in Zenodo [20].

Through field notes, researchers documented nonverbal cues, group interactions, and emerging themes, providing additional layers of context beyond verbal responses.

FGDs were audio-recorded (with participant consent) and transcribed verbatim for thematic analysis. Special attention was given to maintaining anonymity and confidentiality (therefore, during the transcription, all personal information was not included).

## Units of study

Stakeholder groups and thematic areas were selected as units of study to ensure a comprehensive analysis of how the Mental Health Sensitive School Concept (MHSS) can be developed and implemented in the Ukrainian educational system.

**Stakeholder groups (Participants as units of study).** The study engaged multiple key actors within the school system to gather diverse perspectives on mental health-sensitive practices. The primary stakeholder groups included school administrators, teachers, psychologists, social pedagogues, and school students (8–11th grades).

**Thematic areas (Conceptual units of study).** The study examined core thematic areas related to mental health in schools, structured around key components of the MHSS. These thematic areas included Mental Health Awareness and Stigma Reduction, School-Based Mental Health Interventions, Policy Development and Institutional Support, Target Groups and Accessibility of Services: Teacher and Staff Capacity-Building, Adaptation to Crisis Contexts, Models, and Best Practices.

## Data processing

Data processing involves several stages, including transcription, coding, and thematic analysis. All focus groups, conducted across multiple regions of Ukraine, were audio-recorded (with participant consent) and later transcribed verbatim. This ensured that participants' spoken dialogue, tone, and key expressions were preserved, allowing for accurate representation of stakeholder views. To enhance data fidelity, transcripts were cross-checked with field notes taken during the discussions to capture non-verbal cues, group dynamics, and contextual factors.

## Data analysis

Thematic analysis was applied to examine the data collected from focus group discussions, ensuring a structured yet flexible approach to identifying key insights. A combination of inductive and deductive approaches was utilised. An inductive coding process initially allowed themes to emerge naturally from participants' discussions. Following the inductive phase, a deductive coding process was applied. Data analysis was conducted considering participants' subjective experiences, discussion power dynamics, and the practical implications for implementing and scaling mental health-sensitive interventions in schools. The iterative nature of thematic analysis allowed for continuous refinement of categories and validation of insights, ensuring that the final themes accurately captured stakeholder perspectives and actionable recommendations for improving school-based mental health support.

## Techniques to enhance trustworthiness

A diverse range of participants (e.g., teachers, students, psychologists, social pedagogues, and school administrators) was included to capture a broad spectrum of experiences and perspectives, increasing the potential for findings to be

relevant in various school contexts. Detailed notes from meetings and recorded FGDs were reviewed for consistency. Data were cross-verified through audio records, transcripts, and observational meeting notes. Preliminary themes and interpretations were shared with selected participants to confirm whether they accurately reflected their perspectives and experiences. All three researchers conducted thematic analysis, and coding consistency was ensured through independent coding and comparison of results to enhance objectivity. Regular discussions between researchers were conducted to review coding decisions, challenge assumptions, and refine interpretations to prevent individual researcher biases from shaping the findings. Researchers acknowledged assumptions, biases, and power dynamics to minimize influence on findings.

## Results

### Distribution of participants by roles and location

Overall, 12 focus groups (2 groups each on August 15, 16, 30, 31, September 8, and 9, 2022) were conducted. The contingency table with the distribution of focus group participants according to their role in the education system, their regional affiliation, and all other underlying data is available in Zenodo [20]. A total of 205 participants were involved in the study, representing various roles within the school community across six regions in Ukraine (Lviv, Dnipropetrovsk, Rivne, Chernivtsi, Kyivska) and a small category labeled "No data," indicating respondents whose regional information was not recorded.

The most significant number of participants came from Rivne oblast (147 participants, 71.7% of the total), indicating significant engagement from this region. Lviv (27 participants, 13.2%) and Dnipropetrovsk oblasts (12 participants, 5.9%) had moderate representation. Chernivtsi (10 participants, 4.9%) and Kyivska oblasts (1 participant, 0.5%) had the lowest number of respondents.

Teachers (77 participants, 37.6%) formed the largest group (58 teachers). School administrators (40 participants, 19.5%) were the second-largest group from Rivne (26). Students were the third largest group (38 participants, 18.5%). School psychologists (26 participants, 12.7%) were present across all regions. 17 participants, 8.3%, presented social pedagogues. Teacher-organisers (5 participants, 2.4%), teacher's assistants (1 participant, 0.5%) and nurses (1 participant, 0.5%) had minimal representation. Eight participants (3.9%) did not have their regional affiliation recorded. These included two students, three school administrators, two psychologists, and one teacher.

### Perceptions of MHSS by key stakeholder groups

**Teachers.** Teachers generally felt that mental health issues should be primarily handled by psychologists, but acknowledged the importance of having a basic understanding of mental health challenges to better support students. Many emphasised the need for their own mental health support, particularly in managing burnout and stress. Some proposed mandatory psychological training for educators and suggested incorporating mental health awareness into school charters and policies. A teacher commented: *"We see psychological problems, but we are more focused on our work, on teaching our subject."*

**School psychologists and social pedagogues.** Psychologists and social pedagogues recognised the importance of the MHSS yet expressed concerns about their already heavy workloads, questioning how additional responsibilities could be accommodated. They stressed the need for methodological support and clear guidance on their roles in implementing the concept. Some voiced doubts about the feasibility of implementation without external motivation and systemic changes. One psychologist explained: *"If a psychologist wanted to improve their skills, I would go to a methodological office and be shown how to implement this concept, and then I would come and implement it myself at school. Where I am alone, it is ineffective."*

**School administrators.** School administrators largely supported mental health support for teachers and suggested expanding the number of psychologists per school, as one psychologist per institution was deemed insufficient. They also highlighted challenges related to school autonomy and the necessity of governmental support for the MHSS's integration

---

into school structures. An administrator remarked: *"First and foremost, there should be education for parents and teachers, seminars and trainings conducted by psychologists."*

**Students.** Students identified bullying, academic overload, and lack of mental health support as significant concerns. Many reported not taking existing school psychological services seriously, citing a lack of engagement and trust in school psychologists. Some students also mentioned experiencing teacher favouritism, which contributed to classroom inequality. Overall, students supported the idea of mental health-sensitive schools but emphasised the need for qualified, accessible, and confidential mental health support. One student noted: *"There is no school psychologist; we do not know where to turn."*

In general, the focus group discussions revealed a broad consensus on the relevance of integrating mental health support into schools, particularly in the context of war and prolonged stress. Participants acknowledged the importance of the Mental Health Sensitive School Concept but expressed concerns regarding its implementation, particularly in terms of resource availability, staff training, and policy integration.

### Core thematic areas related to mental health in schools

The focus group discussions revealed a detailed understanding of various aspects of the Mental Health Sensitive School Concept, which were categorised into six key areas: Mental Health Awareness and Stigma Reduction, School-Based Mental Health Interventions, Policy Development and Institutional Support, Target Groups and Accessibility of Services, Teacher and Staff Capacity-Building, and Adaptation to Crisis Contexts, Models, and Best Practices.

**Mental health awareness and stigma reduction.** Participants across all stakeholder groups emphasised the importance of fostering mental health awareness within school environments and addressing the persistent stigma associated with psychological distress. Students, in particular, reported hesitance in approaching school psychologists due to concerns about confidentiality and the perceived ineffectiveness of school-based interventions. One student noted: *"There is no school psychologist; we do not know where to turn."*

Teachers and school administrators acknowledged the need for structured awareness programs for students and staff to normalise conversations around mental health. A participant suggested: *"Convincing children that coming to a psychologist's office is not scary is crucial; they just need to talk."* Teachers also pointed out the role of parents in fostering mental health awareness, recommending the development of parent engagement programs to ensure continuity between home and school environments.

**School-based mental health interventions.** The effectiveness of mental health interventions within schools was a central theme. Many schools reported implementing anti-bullying campaigns, resilience training, and stress management programs, but the perceived effectiveness of these interventions varied. Some teachers pointed out that existing programs lack structured methodologies and ongoing evaluation. One teacher expressed: *"We have programs, but they seem to exist only on paper and have no real impact on students."*

Psychologists and social workers stressed the importance of integrating social-emotional learning (SEL) practices into curricula rather than offering mental health programs as one-time activities. One psychologist commented, *"At the school level, we do promotion and prevention, but structured integration is missing."* Moreover, peer-to-peer support models were highlighted as a practical approach, with students suggesting that high school students could be trained to support younger peers in managing stress and emotional difficulties.

**Policy development and institutional support.** The need for formal policy frameworks to support the MHSSC was widely discussed. While some schools embedded mental health policies in their charters or development strategies, others lacked formal documentation guiding mental health initiatives. An administrator pointed out: *"Each school will not develop its own; it must come from a centralised source."*

There was a strong call for governmental endorsement and integration of mental health-sensitive policies within national educational frameworks. Administrators suggested mandatory teacher training on mental health and allocating

additional staff at the school level. One school representative emphasised: *"Without formal policy backing, these initiatives may remain fragmented and inconsistently implemented across schools."*

**Target groups and accessibility of services.** Identifying target groups for interventions emerged as a contentious issue. Teachers and psychologists raised concerns about defining the boundaries between different levels of need, especially in light of the ongoing crisis. One teacher asked, *"Who does not have mental health risks now? Wherever you live, everyone is affected in some way."*

Students also voiced concerns about stigmatisation and potential bullying associated with identifying specific target groups for mental health interventions. One student noted: *"How will these groupings be kept confidential? I would not want my teacher to know what group I belong to."* Psychologists emphasised that any classification must be done sensitively and accompanied by comprehensive training for staff on how to manage mental health disclosures appropriately.

**Teacher and staff capacity-building.** The mental health of teachers and school staff was frequently mentioned as a priority area for intervention. Many teachers expressed concerns about burnout and emotional fatigue, with some noting that mental health training should be a mandatory component of professional development programs. A teacher stated: *"Teachers go for advanced training every five years, but there should be mental health training to avoid burnout."*

Additionally, the importance of peer support networks for teachers was highlighted, with suggestions that schools establish supervisory groups where educators can discuss mental health challenges openly. One participant remarked: *"Teachers need their own support networks, just as students do."*

**Adaptation to crisis contexts, models, and best practices.** Given the current crisis, there was a significant focus on adapting mental health-sensitive school models to war-affected contexts. Participants emphasised the need for trauma-informed practices, structured stress-reduction techniques, and guidance for supporting displaced students (IDPs). A teacher shared: *"I would develop materials for students and parents in case of an air raid - how to act, reduce panic, and remain stress-resistant."*

Several participants pointed out that existing international models could be adapted to Ukraine's specific needs. Examples of well-being tracking systems from Israel and peer-supported mental health models from the UK were discussed. A school psychologist emphasised: *"If we want this to work, we need structured guidance on best practices, not just theoretical frameworks."*

## Implementation: Challenges and steps forward

Participants identified several challenges related to the implementation of the MHSS. Among them were:

- Lack of clarity on policy development and ownership: Many stakeholders were uncertain about who should develop and enforce mental health policies within schools. Some suggested integrating mental health initiatives into school development strategies or formalising them through ministerial programs. A participant stated: *"It can be a program, like the Patriotic Education Program, that comes down from the top of the Ministry of Education and Science, with everything written down by year, responsible units, and feedback."*

- Limited availability of psychologists and mental health professionals: The current number of school psychologists was considered insufficient, particularly in large schools. One psychologist expressed: *"We have 1,500 children, a large school. We have one psychologist and one social worker. It is difficult to cover everyone."*

- Parental involvement: Teachers and psychologists emphasise that parents play a critical role in children's mental health, but engaging parents in mental health initiatives remains a challenge. A psychologist shared: *"Without parents, everything falls through. If parents are involved, then this component works."*

- Resistance to change: Some teachers and school administrators expressed concerns that mental health policies might remain paper-based formalities rather than leading to tangible improvements in school environments. One teacher remarked: *"We need something more down-to-earth."*

Participants suggested several practical steps to facilitate the successful implementation of the MHSS:

- Training for Educators and School Staff: Schools should offer mandatory training on mental health first aid, stress management, and trauma-informed practices. One participant noted, *"First of all, it is a teacher. Teach teachers to understand their condition. Training."*

- Strengthening Mental Health Support System: Increase the number of school psychologists, integrate peer-to-peer support programs, and provide additional funding for mental health services.

- Student-Centered Mental Health Education: Implement interactive workshops, mental health awareness campaigns, and confidential consultation services for students. One student suggested introducing *team-building into the curriculum of younger classes to counteract bullying, unite students, and make friends.*

- Parental Engagement: Introduce parenting workshops and develop collaborative programs that support family involvement in students' mental well-being. One participant emphasised: *"The child transfers everything in the family to school. They should be directed to work with parents."*

- Policy Integration: Secure governmental approval for the MHSSC, ensuring its alignment with national education policies.

## Discussion

The focus group discussions findings provide valuable insights into integrating mental health-sensitive practices in schools. The high representation of teachers and school administrators highlights their central role in shaping school-based mental health interventions. The relatively low participation of students (18.5%) indicates that further efforts may be needed to increase student engagement in discussions on mental health-sensitive school policies. The involvement of school psychologists (12.7%) and social pedagogues (8.3%) underscores the importance of specialist mental health roles in implementing the MHSS. The underrepresentation of school nurses and teachers' assistants suggests that these roles may be less involved in school mental health initiatives or that their participation was not actively sought.

Participants were not simply offering abstract reflections on mental health in schools; they were speaking from within a setting of displacement, grief, uncertainty, chronic stress, and rapidly shifting institutional demands. These conditions led to an **increased urgency** in demands for school-based mental health interventions and clearer referral pathways; greater **emphasis on trauma-informed approaches**, including psychological first aid and protective school climates; a heightened awareness of the **emotional toll on educators**, who are navigating dual responsibilities as caregivers and victims of conflict; calls for **adaptability and resilience** in MHSS design, with participants favouring flexible, scalable interventions over rigid programmatic models. This wartime context both sharpened stakeholders' insights and added emotional weight to their contributions. It also affected feasibility assumptions, as participants often framed their input in terms of *what could realistically be done now*, given resource scarcities, rather than idealised future models.

While there was a broad consensus on the need to integrate mental health into school environments, stigma and lack of awareness remain barriers to student engagement. The hesitation of students to approach school psychologists due to concerns about confidentiality and perceived ineffectiveness suggests the need for awareness campaigns and peer-led mental health initiatives.

Existing interventions were acknowledged but often deemed insufficient. Teachers and psychologists expressed a need for structured, evidence-based programs that go beyond sporadic anti-bullying or stress management sessions. They strongly emphasised training teachers and psychologists to handle student disclosures sensitively and confidentially. Teachers and administrators also called for trauma-informed practices and the adaptation of international school mental health models to fit the Ukrainian educational system.

The burnout risk among teachers and school psychologists was a recurring theme. The findings indicate a pressing need for systematic mental health support for educators, including self-care programs and peer-support networks.

Regarding implementation, stakeholders stressed that mental health policies should be endorsed centrally rather than developed independently by schools. The request for ministerial-level guidance reflects a structural gap in mental health governance in schools.

The findings align with global studies on school-based mental health programs, emphasising the effectiveness of whole-school approaches [21,22]. The concerns about mental health accessibility and the need for mental health integration into education reflect broader trends in mental health research, particularly in post-conflict and trauma-affected societies [23,24]. The need for policy endorsement aligns with international frameworks, such as the World Health Organization's (WHO) recommendations for promoting health in education systems [25].

However, while stakeholder perspectives informed the framework design, we acknowledge that these perspectives are embedded within broader discourses and may carry unexamined assumptions. For instance, the emphasis on mental health awareness and stigma reduction, while widely supported in global literature, has shown mixed outcomes depending on implementation context. Some studies suggest that awareness campaigns may unintentionally increase stigma by reinforcing illness-based framing, particularly when lacking complementary behaviour change strategies [26,27]. Therefore, future MHSS iterations should critically evaluate which forms of awareness-raising are most effective and culturally safe in specific Ukrainian school settings. Likewise, while participants advocated strongly for capacity-building, literature from conflict-affected settings cautions against over-reliance on non-specialist-delivered psychological support without adequate supervision or referral systems [28]. The MHSS addresses this by delineating roles and ensuring stepped-care alignment, but continuous monitoring will be required to ensure that role expansion does not lead to burden or mission drift.

Overall, integrating the participatory findings into the MHSS framework is a strength, but future implementation must retain space for critical reflection and adaptation as both evidence and context evolve.

Based on the results of FGDs, which provided crucial insights, the MHSS Concept was finalised. **Mental Health Awareness and Stigma Reduction** theme informed the integration of universal mental health promotion activities in **Component 3: Intervention Areas** and supported the need for student-led campaigns and inclusive school environments in **Component 4: Key Interventions**. **School-Based Mental Health Interventions** theme shaped the tiered approach outlined in **Component 2: Target Groups**, which distinguishes between universal, at-risk, and referral-level students, and informed intervention selection within **Component 3**. **Policy Development and Institutional Support** theme provided the rationale for **Component 1: Policy**, embedding mental health strategies into official school development plans and guiding the formation of Mental Health Task Forces. T**arget Groups and Accessibility of Services** theme strengthened stratification logic in **Component 2**, including mechanisms for identifying and supporting students with varying levels of need, and ensuring accessibility regardless of socioeconomic or geographic background. **Teacher and Staff Capacity-Building** theme Directly shaped **Component 5: Key Actors**, which outlines training and support structures tailored to the roles of teachers, administrators, psychologists, and other school staff. **Adaptation to Crisis Contexts, Models, and Best Practices** theme influenced the inclusion of trauma-informed, context-responsive strategies across all components, and ensured alignment with global frameworks such as WHO's Helping Adolescents Thrive [11] and IASC's MHPSS [29].

**Five key principles** guide the current MHSS framework. First, it follows a systemic approach, embedding mental health considerations into school policies and daily practices. Second, it adopts a needs-centered focus, ensuring interventions are tailored to different groups' specific mental health requirements. Third, it incorporates a tiered support model, offering a range of interventions from universal awareness campaigns to targeted mental health support and specialised care referrals. Fourth, the concept prioritises evidence-based practices, ensuring that all implemented programs are supported by scientific research. Finally, the framework is competency-driven, meaning that school staff, including teachers and psychologists, must be adequately trained to address mental health issues effectively.

Structurally, the MHSS consists of **five core components**, forming a mental health support pyramid (Fig 1). The first component focuses on developing school mental health policies that integrate mental health awareness and interventions into school governance. The second component identifies and categorises target groups within the school community, ensuring tailored support for students and staff based on their mental health needs. These groups include individuals who require general well-being promotion, those at risk of psychological distress, those in need of active mental health support, and those requiring specialised professional intervention.

The third component outlines the areas of intervention, covering mental health promotion, preventive measures, early mental health support, and structured referral mechanisms. The fourth component describes the key interventions, which include teacher training, student support programs, structured psychological interventions, and the development of safe school environments that promote emotional regulation and mental well-being. The final component defines the key actors responsible for implementing the MHSS, including school administrators, teachers, psychologists, social workers, parents, and community partners. Each group plays a crucial role in fostering a mentally healthy school environment through collaboration and active engagement.

The MHSS provides a **step-by-step roadmap** for schools to follow to support implementation. The process begins with establishing a Mental Health Task Force and developing a school-wide mental health policy that aligns with national education strategies. Schools then conduct an assessment to identify the mental health needs of students and staff, which informs the selection and implementation of tiered interventions, ranging from mental health promotion to targeted mental health support and specialist referrals. Finally, regular monitoring and evaluation ensure that interventions remain effective and responsive to the evolving needs of the school community.

The Mental Health Sensitive School Concept represents a proactive and holistic approach to integrating mental health into educational settings. Its structured yet adaptable nature ensures that mental health is recognised not as a separate issue but as a fundamental pillar of holistic education, reinforcing the idea that there is no health without mental health.

The study reinforces the need for a nationally endorsed MHSS policy integrated into Ukraine's broader education reform. Teacher training should include mental health literacy, psychological first aid, and self-care strategies to address stress and burnout. Schools should move from reactive interventions (addressing mental health issues once they arise) to preventive models, incorporating mental well-being into daily school activities. Given the current war context, interventions should incorporate mental health support mechanisms for both students and educators, including specialised trauma-focused programs.

While the MHSS framework was designed in response to the current mental health needs that existed before the war and were exacerbated during it, it can also serve as a foundation for post-conflict recovery and resilience. The psychological consequences of war, such as chronic stress, unresolved grief, disrupted attachment, and intergenerational trauma, will likely reverberate for many years. In this context, the MHSS cannot treat the war merely as a temporary disruption but must institutionalise long-term capacity for trauma-informed education and restoration. Embedding the war's ongoing legacy into the MHSS pyramid explicitly could ensure that schools are not only sites of immediate protection but also engines of societal healing.

## Conclusions

The FGDs highlighted a strong demand for integrating mental health-sensitive approaches into schools while identifying critical barriers that must be addressed. Participants widely supported the concept but stressed the need for concrete steps, systemic changes, and active involvement of all stakeholders. The six thematic areas identified provide a structured roadmap for developing and refining the Mental Health Sensitive School Concept. The findings indicate that the successful implementation of the MHSSC will require policy backing, resource allocation, and structured capacity-building programs at all levels of the education system. A psychologist from the FGD summarised: "*The reality is that our society will need this - whether within this framework or another - it has to be implemented.*"

## Limitations of the study

This study has several limitations. While Rivne oblast had the highest participation, other regions had lower engagement, which may limit the generalizability of findings across Ukraine (as well as sampling constraints). Given the focus group methodology, responses may have been influenced by social desirability bias, particularly among teachers and school administrators. Also, we recognise research biases (intervention-optimist and cultural embeddedness biases). The study does not assess the long-term impact of implementing the MHSS, highlighting the need for future implementation studies on the Mental Health Sensitive School Concept.

## Author contributions

**Conceptualization:** Vitalii Klymchuk.

**Data curation:** Olha Faryma.

**Formal analysis:** Olha Faryma.

**Methodology:** Viktoriia Gorbunova, Olha Faryma.

**Writing – original draft:** Vitalii Klymchuk, Viktoriia Gorbunova.

**Writing – review & editing:** Viktoriia Gorbunova, Olha Faryma.

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
