## [Decision Letter · Decision Letter 0]

PMEN-D-25-00099

Developing the Mental Health Sensitive School Concept: Insights from Focus Group Discussions

PLOS Mental Health

Dear Dr. Klymchuk,

Thank you for submitting your manuscript to PLOS Mental Health and my apologies for the delay. After careful consideration of the reviewer reports, we feel that your paper has merit but does not yet fully meet PLOS Mental Health’s publication criteria as it currently stands. Therefore, we invite you to submit a revised version of the manuscript that addresses the points raised during the review process.

Please address all of the comments raised by the reviewers, which you can find at the end of this email and attached. Please ensure that you pay careful attention to the comments raised by reviewer 1 in particular.

We look forward to receiving your revised manuscript.

Kind regards,

Karli Montague-Cardoso

Executive Editor

PLOS Mental Health

Journal Requirements:

1. We have amended your Competing Interest statement to comply with journal style. We kindly ask that you double check the statement and let us know if anything is incorrect.

2. Please upload a copy of Figure 1 which you refer to in your text on page 15. Or, if the figure is no longer to be included as part of the submission please remove all reference to it within the text.

3. Please provide separate figure files in .tif or .eps format.

https://journals.plos.org/mentalhealth/s/figures 

https://journals.plos.org/mentalhealth/s/figures#loc-file-requirements

4. We have noticed that you have uploaded Supporting Information files, but you have not included a list of legends. Please add a full list of legends for your Supporting Information files after the references list.

Additional Editor Comments (if provided):

Reviewers' comments:

Reviewer's Responses to Questions

**Comments to the Author**

1. Does this manuscript meet PLOS Mental Health’s publication criteria?

Reviewer #1: Partly

Reviewer #2: Yes

2. Has the statistical analysis been performed appropriately and rigorously?

Reviewer #1: N/A

Reviewer #2: Yes

3. Have the authors made all data underlying the findings in their manuscript fully available (please refer to the Data Availability Statement at the start of the manuscript PDF file)?

Reviewer #1: Yes

Reviewer #2: Yes

4. Is the manuscript presented in an intelligible fashion and written in standard English?

Reviewer #1: Yes

Reviewer #2: Yes

Reviewer #1: Developing the Mental Health Sensitive School Concept: Insights from Focus Group Discussions

This study offers a qualitative assessment of a Mental-Health-Sensitive School (MHSS) Concept with special attention to lived experiences by pupils and school staff. It is exemplary and commendable that the authors have sought input from pupils and school staff—key stakeholders in the school context. This represents an essential, yet frequently overlooked, step in the development of meaningful, contextually appropriate, and acceptable MHPSS frameworks.

The introduction presents a rationale for the study, outlines the key research goals, aims and objectives. The description of the MHSS framework remains somewhat vague to the reader and (1) a more thorough description on what the model concretely entails would benefit the manuscript. Relatedly, (2) key terms should be defined: terms such as psychological support, mental health support, mental health and psychosocial support, mental health, psychosocial support are used throughout the manuscript and also by participants. It is unclear for the reader what different levels of support the terms represent or if the terms are used interchangeably. Do some of the terms refer for example to clinical psychological care for mental disorders, and other to psychosocial support to manage everyday stress?

Throughout the manuscript, claims are made that (3) should be linked to earlier research (references).

The methodology is appropriate. The choice of thematic analysis aligns well with the research questions. (4) The sampling techniques should be described in greater detail, for example how participants were recruited. I applaud the authors’ attention to power dynamics, safety and respect for the participants, however, in relation to the overall manuscript these descriptions seem lengthy.

There are three authors to the paper, however, the methods section mentions that “multiple” researchers conducted the thematic analysis. (5) I recommend describing somewhat more in detail the numbers and backgrounds of the researchers, although not necessarily to an individual level. The awareness of assumptions and bias is mentioned, but bias are not critically discussed and analyzed, neither in the limitation section (6).

Key themes were identified—mental health awareness and stigma reduction, school-based interventions, policy development, capacity-building, and adaptation to crisis contexts. The voice of the participants is represented well (e.g. through quotes). The described interpretative process is generally sound.

Comments regarding the discussion (7): The identified themes cover the field of MHPSS comprehensively and, perhaps given the constraints of space, the discussion of each of them remains superficial. It is unclear for the reader how the MHSS framework was developed based on the FGD input given that the original framework was not described in detail. The description of the finalized MHSS framework remains somewhat unattached to the input from the FGDs. Further, the input is not critically discussed in light of existing literature; for example, there are findings indicating that mental health awareness may be counterproductive in certain conditions.

Minor: p. 11 The manuscript says that special attention was given to anonymity, however, the FGD is said to have started with introductions.

Thank you for the opportunity to review this important work. It is pleasing to see that MHPSS models are being created and developed in collaboration with and with respect to the ones who ultimately are the targets for these initiatives.

Recommendation: Major revision

Reviewer #2: Nice study, well written. I feel like perhaps there is not enough touching on the impacts of the ongoing conflict and how this influences every stakeholders' answers to the focus groups?

Ln 160 - you have "to balance these perspectives" twice in the same sentence.

Results

Ln 409 - adaptation to crisis context. I understand that being ready to react adequately in the context of an attack is paramount. However, I wonder whether there is space/room for children and teachers to simply talk about how difficult it is to be facing war (and to have been for so long?). Do they have opportunities to explore existential issues? To talk about the impact the war has had on their mental health?

Discussion

-Is there space for the war (addressing its effect, talking about it, etc.) in the MHSS pyramid? It seems like the impacts of this conflict will continue to reverberate for many years, possibly decades (even once the conflit is over), perhaps it should be added in there?

**Do you want your identity to be public for this peer review?** For information about this choice, including consent withdrawal, please see our Privacy Policy

Reviewer #1: No

Reviewer #2: **Yes: ** Catherine Malboeuf-Hurtubise

---

## [Editor Report · Decision Letter 1]

Developing the Mental Health Sensitive School Concept: Insights from Focus Group Discussions

PMEN-D-25-00099R1

Dear Dr. Klymchuk,

We are pleased to inform you that your manuscript 'Developing the Mental Health Sensitive School Concept: Insights from Focus Group Discussions' has been provisionally accepted for publication in PLOS Mental Health.

Best regards,

Karli Montague-Cardoso

Executve Editor

PLOS Mental Health